# Flavonoid Mixture Inhibits *Mycobacterium tuberculosis* Survival and Infectivity

**DOI:** 10.3390/molecules24050851

**Published:** 2019-02-28

**Authors:** Ruoqiong Cao, Garrett Teskey, Hicret Islamoglu, Myra Gutierrez, Oscar Salaiz, Shalok Munjal, Marcel P. Fraix, Airani Sathananthan, David C. Nieman, Vishwanath Venketaraman

**Affiliations:** 1Department of Basic Medical Sciences, College of Osteopathic Medicine of the Pacific, Western University of Health Sciences, Pomona, CA 91766-1854, USA; rcao@westernu.edu (R.C.); gteskey@westernu.edu (G.T.); myra.gutierrez@westernu.edu (M.G.); oscar.salaiz@westernu.edu (O.S.); shalok.munjal@westernu.edu (S.M.); 2Western University of Health Sciences College of Dental Medicine, Pomona, CA 91766-1854, USA; hislamoglu@westernu.edu; 3Departments of Physical Medicine and Rehabilitation and Neuromusculoskeletal Medicine/Osteopathic Manipulative Medicine, Western University of Health Sciences College of Osteopathic Medicine of the Pacific, Pomona, CA 91766-1854, USA; mfraix@westernu.edu; 4Department of Internal Medicine, Western University of Health Sciences College of Osteopathic Medicine of the Pacific, Pomona, CA 91766-1854, USA; asathananthan@westernu.edu; 5Department of Health and Exercise Science, Appalachian State University, North Carolina Research Campus, Kannapolis, NC 28081, USA; niemandcasu@gmail.com

**Keywords:** *Mycobacterium tuberculosis*, *Mtb*, TB, flavonoids, immunomodulation, glutathione, T2DM, polyphenols

## Abstract

Background: Flavonoids have been shown to exert anti-pathogenic potential, but few studies have investigated their effects on *Mycobacterium tuberculosis* (*Mtb*) infectivity. We hypothesized that a flavonoid mixture would have a favorable influence on cell death and the resolution of *Mtb* infection in THP-1 macrophages and in granulomas derived from both healthy participants and those with type 2 diabetes mellitus (T2DM). METHODS: THP-1 macrophages, and in vitro granulomas from healthy participants (*N* = 8) and individuals with T2DM (*N* = 5) were infected with *Mtb*. A mixed flavonoid supplement (MFS) at a concentration of 0.69 mg per ml was added as treatment to *Mtb* infected THP-1 macrophages and granulomas for 8 to 15 days. RESULTS: MFS treatment significantly reduced the intracellular *Mtb* survival, increased cell density, aggregation, and granuloma formation, and increased glutathione (GSH) levels. IL-12 and IFN-γ levels tended to be higher and IL-10 lower when *Mtb* infected THP-1 macrophages and granulomas obtained from healthy subjects were treated with MFS compared to control. CONCLUSIONS: MFS treatment exerted a strong influence against *Mtb* infectivity in THP-1 macrophages and in granulomas including antimycobacterial effects, GSH enrichment, cytokine regulation, and augmented granuloma formation. Our data support the strategy of increased flavonoid intake for managing tuberculosis.

## 1. Introduction

*Mycobacterium tuberculosis* (*Mtb*), the etiological agent which causes tuberculosis (TB), is the leading cause of infectious death worldwide. *Mtb* is transmitted from person to person via aerosol droplets typically by means of coughing. In 2017, there was an estimated 10.0 million new cases of TB reported, and 1.3 million deaths worldwide among human immunodeficiency virus (HIV)-negative individuals [1]. Typical treatment of active TB includes an extensive antibiotic regimen of isoniazid, rifampin, ethambutol and pyrazinamide, which have specific mechanisms of action in combating *Mtb* infection. However, due to the emergence of antibiotic-resistant strains of *Mtb* treatment is becoming more problematic [2]. Therefore, many new treatment options are being explored as possible alternatives or adjunctive therapies to combat this devastating disease. There is currently an urgency to find new treatment procedures for *Mtb* infections due to the emergence of multi-drug resistant strains of *Mtb* [1].

Despite substantial advancements attended in this field, diabetic patients are still considerably more likely to contract TB than nondiabetic individuals [3]. The number of people with diabetes has risen from 108 million in 1980 to 422 million in 2014. Four out of five adults with diabetes live in low- and middle-income countries where the prevalence of TB and other infectious diseases is high. In 2016, an estimated 1.6 million deaths were directly caused by diabetes [4]. It is commonly accepted that diabetes decreases the effectiveness of cell mediated immunity thus making these individuals more vulnerable to *Mtb* infection, but the exact mechanism for *Mtb* susceptibility in individuals with T2DM is not entirely well understood [3]. 

Polyphenols are a structural class of organic plant-based compounds, characterized by the presence of multiple units and responsible for many natural food pigmentations. Nearly half of the polyphenols are flavonoids, which can be further divided into flavan-3-ols, flavanones, flavones, isoflavones, flavonols, anthocyanins and proanthocyanidins [5]. The Phenol-Explorer database on polyphenol content in foods contains values for 500 different polyphenols in over 400 foods in the human diet (http://phenol-explorer.eu/). Recommendations for dietary polyphenol and flavonoid intake have not yet been established. In Europe, the average dietary polyphenol intake has been estimated at 1.187 g/day with coffee, tea, fruits, and wine as the principal sources [6]. Nearly all ingested polyphenols pass through the small intestine unabsorbed and reach the colon where bacterial degradation produces smaller phenolics that can be reabsorbed into the circulation after undergoing phase 2 conjugation in the liver [7]. The gut-derived phenolics circulate throughout the body exerting a variety of bioactive effects. Studies support a robust relationship between high dietary polyphenol intake and reduced risks for overall mortality and several chronic diseases, acute respiratory illness, inflammation, and oxidative stress [7,8,9]. Two human studies suggest an inverse relationship between regular intake of flavonoid-rich tea beverages and risk for TB [10,11].

Cell culture studies support a strong anti-pathogenic influence from selected flavonoids including quercetin, anthocyanins, and flavan-3-ols [12,13,14]. For example, serum samples collected from athletes that contained metabolites from blueberry and green tea ingestion protected cells from killing by the vesicular stomatitis virus [12]. Limited evidence indicates that epigallocatechin-3-gallate (EGCG) and quercetin inhibit *Mtb* growth and survival within human macrophages [13,14]. We sought to extend these findings by measuring the influence of a flavonoid mixture (quercetin, green tea flavan-3-ols, bilberry anthocyanins) on cell death and resolution of infection in a variety of *Mtb* infected cells. 

## 2. Results

### 2.1. Survival of Mtb Erdman Subsequent to MFS Treatment

The direct antimycobacterial effects of the MFS was determined during the log phase *Mtb* growth in 7H9 media. After eight days of incubation, a twofold decrease in bacterial quantity was measured when *Mtb* was treated with the MFS compared to sham treatment (Figure 1).

### 2.2. Hematoxylin and Eosin Staining of THP-1 Cells

Hematoxylin and Eosin (H&E) staining of THP-1 macrophages showed an increase in cell quantity after MFS treatment versus the sham-treated after 12 days of infection (Figure 2). These data indicate that THP-1 macrophage viability was enhanced after MFS treatment. 

### 2.3. Intracellular Survival of Mtb Erdman from THP-1 Cells

The bacterial quantity of *Mtb* Erdman was approximately 2.5 times lower when THP-1 cells were treated with MFS versus no treatment after 12 days (Figure 3). These data indicate that MFS treated THP-1 cells exhibited an enhanced ability to eliminate and/or contain the *Mtb* infection. 

### 2.4. Levels of IL-10 from THP-1 cell Supernatant

Although not statistically significant, IL-10 levels were diminished when THP-1 cells were treated with MFS compared to the sham-treated infected-control (Figure 4).

### 2.5. Levels of Total GSH from Mtb Erdman Infected THP-1 cell Lysate

*Mtb* infected THP-1 cells displayed a statistically significant increase in GSH levels after MFS treatment compared to the control category (Figure 5).

### 2.6. Hematoxylin and Eosin Staining of Human Granulomas from Healthy Individuals

H&E staining of *Mtb* infected peripheral blood mononuclear cells (PBMCs) illustrates an increase in cell survival with MFS administration versus sham treatment after 15 days of infection (Figure 2), as well as an increase in cell density/aggregation and granuloma formation (Figure 6). 

### 2.7. Intracellular Survival of Mtb Erdman from the in vitro Granulomas of PBMCs Drawn from Healthy Subjects

The bacterial quantity of *Mtb* Erdman was approximately twofold lower from the granulomas formed from the PBMCs of healthy subjects administered with MFS versus the sham-treated after 15 days post infection (Figure 7).

### 2.8. Levels of Total GSH from Healthy Subjects

*Mtb* infected granulomas from PBMCs of healthy subjects showed a 2.5-fold greater increase in intracellular GSH levels after MFS versus sham treatment (Figure 8). 

### 2.9. Levels of IL-12, IFN-γ, IL-10 from the Supernatants of Healthy Subjects

After 15 days of infection, the IL-12 and IFN-γ levels tended to be higher, and IL-10 lower when *Mtb* infected granulomas derived from PBMCs obtained from healthy subjects were treated with MFS compared to the sham-treated infected-control (Figure 9, Figure 10 and Figure 11). 

### 2.10. Hematoxylin and Eosin Staining of Human Granulomas from Individuals with T2DM

H&E staining of *Mtb* infected granulomas from individuals with T2DM showed an increase in cell density and aggregation with MFS versus sham treatment after 15 days (Figure 12).

### 2.11. Survival of Mtb Erdman Inside Granulomas from Individuals with T2DM

The bacterial quantity of *Mtb* Erdman observed was lower from the granulomas formed from T2DM individuals following MFS administration versus the sham-treated after 15 days (Figure 13). The overall bacterial quantities were substantially higher among the T2DM individuals than observed among the healthy study participants (Figure 14). 

### 2.12. Levels of IL-10 from the Supernatants of Individuals with T2DM

IL-10 levels tended to be lower when *Mtb* infected granulomas from healthy subjects were treated with MFS compared to the sham-treated infected-control after 15 days (Figure 15). Overall, the IL-10 levels in individuals with T2DM were about two-fold higher than observed in the healthy participants (Figure 11).

## 3. Discussion

While TB still plagues roughly one third of the world’s population, flavonoids may offer an exciting and inexpensive alternative therapy due to their direct antimycobacterial and immunomodulating effects. Following food and beverage ingestion, small amounts of flavonoids are absorbed in the small intestine after sulfate, glucuronide, and/or methyl conjugation [7]. Once in the bloodstream, the conjugated flavonoid metabolites can undergo further phase II metabolism in the liver prior to interactions at the cell level and ultimate urinary excretion. Enzymes at the cell level may deconjugate flavonoid metabolites, allowing the more active aglycones to exert beneficial effects as measured in cell culture studies [7]. The majority of ingested flavonoids pass into the large intestine where they undergo extensive microbial catabolism followed by fecal excretion or reentry into the circulatory system as biotransformed phenolic metabolites [7]. Some studies indicate that following flavonoid supplementation, serum samples containing a mixture of conjugated flavonoid metabolites and colon-derived phenolics exert anti-pathogenic effects [12,15]. 

We first measured direct bactericidal influences by adding MFS to *Mtb* suspended in 7H9 broth media. After 15 days (Figure 1), MFS versus control treatment resulted in a reduction in bacterial load (Figure 1). These results demonstrated that flavonoids exerted direct antibacterial action against *Mtb,* although we cannot rule out the potential synergistic effects of the added adjuvants caffeine, vitamin C, and omega 3 fatty acids. This result compares favorably with those from Sasikumar et al. [13] who showed that quercetin exhibited 99% and 56% inhibition against *Mtb* H37Rv at 200 µg/mL and 50 µg/mL, respectively.

Subsequently, we measured the effects of 12 days of MFS treatment on *Mtb* infected human macrophages derived from THP-1 monocytes. MFS supplementation compared to the control condition over a 12-day period improved the viability of the human THP-1 cells as depicted by the H&E imaging, which portrayed a greater cell magnitude (Figure 1). This suggested that *Mtb* infection became too vigorous for the THP-1 cells to retain and secure in the absence of MFS prophylactic treatment. This conclusion is further supported by the data represented in Figure 2, which illustrates that after 12 days MFS treatment resulted in a statistically significant bacterial reduction compared to the sham-treated control. These results support the findings of Anand et al. [14] who showed that the green tea flavonoid EGCG inhibited *Mtb* survival within macrophages. This research group showed that EGCG down-regulated host molecule tryptophan-aspartate containing coat protein (TACO) gene transcription, thus influencing phagosome maturation and the capacity to contain *Mtb.*

Of the three cytokines measured in this study, IL-10 was most effected by MFS treatment with levels half that of the control condition (Figure 4). IL-10 is an immunosuppressive cytokine which down regulates the expression of Th1 cytokines, NF-kB activity, and macrophage costimulatory molecules [16]. Thus, in the context of a *Mtb* infection, IL-10′s inhibition subsequent to MFS administration is considered to be favorable. 

Interestingly, the levels of detectable intracellular GSH were elevated following MFS treatment. Our lab has previously demonstrated that among *Mtb* infected cells, the augmentation of GSH will cause mycobactericidal effects in both a direct and immunomodulating indirect manner through downstream mediators [17,18]. Therefore, MFS treatment of THP-1 cells can diminish *Mtb* infectivity in direct and indirect pathways.

The influence of MFS on *Mtb* infected PBMCs collected from eight healthy individuals was similar to the findings from the THP-1 macrophage assays. The H&E data showed enhanced PBMC cell viability, cell aggregation/density, and enriched granuloma formation after MFS treatment (Figure 6). This is significant because augmented granuloma development is linked to increased bacterial containment and improved host protection [19]. As with the THP-1 macrophages, the increased immune cell viability observed subsequent to MFS treatment coincided with a strong reduction in bacterial density among the infected PBMCs (Figure 2 and Figure 6). Taken together with the GSH and cytokine data, these results support direct MFS-related mycobactericidal effects and immunomodulation from granuloma formation and the trend for increased IL-12 and IFN-γ cytokine production (Figure 8, Figure 9, Figure 10 and Figure 11). Both IL-12 and IFN-γ are prominently involved in immune cell differentiation and activation in response to combating *Mtb* infection, a process linked to granuloma formation [20]. 

Lastly, we tested the influence of MFS treatment on PBMCs collected from individuals with T2DM, a patient population that is significantly more susceptible to *Mtb* infections [1]. Consistent with findings from the healthy individuals, PBMCs of T2DM patients treated with MFS exhibited more robust cellular aggregations and granuloma formation (Figure 12). Correspondingly, after MFS administration the PBMC bacterial burden was reduced to roughly one-third that of the T2DM infected control (Figure 13). Although the abundance of *Mtb* was significantly diminished with MFS treatment, the quantity of bacteria remaining was still over double that of levels linked to the same experiments in healthy study participants (Figure 14). This demonstrates that while MFS supplementation reduced *Mtb* infectivity in both healthy individuals and those with T2DM, it is not sufficient to cause complete *Mtb* elimination. 

## 4. Materials and Methods

### 4.1. THP-1 Cell Culture

The THP-1 cell line originated from the American Type Culture Collective (ATCC), was cultured in Roswell Park Institute (RPMI) medium (Sigma, St. Louis, MO, USA) with 10% Fetal bovine serum (FBS-Sigma) and incubated at 37 °C with 5% CO_2_. After 7–12 days, cells were collected from the flasks, spun at 2000 rpm for 15 min, resuspended in RPMI containing 10% FBS and counted for cell numbers. THP-1 cells (2 × 10^5^/well) were distributed in 0.001% poly-lysine (Sigma, St. Louis, MO, USA) coated 24-well tissue culture plates and incubated overnight. Differentiation of THP-1 cells to macrophages was achieved by adding PMA (Phorbol 12-myristate 13-acetate-Sigma) at a concentration of 10 ng/mL.

### 4.2. Culture of Mtb Erdmann

*Mtb* Erdmann expressing green fluorescent protein (GFP) was obtained as a gift from Dr. Selvakumar Subbian at Rutgers New Jersey Medical School, Biomedical and Health Sciences. *Mtb* Erdmann is similar to H37Rv (standard laboratory strain of *Mtb*) however it is considered to be more virulent due to its faster doubling time [21]. *Mtb* Erdmann was used for all experiments conducted in this study, handled inside the biosafety level 3 facility (BSL-3) and cultured in Difco Middlebrook 7H9 broth media supplemented with ADC (Albumin Dextrose Catalase) at 37 °C. Mtb Erdman was processed for infection once the static culture was at the peak logarithmic phase of growth (optical density at 600 nm, between 0.5 and 0.8) and subsequently washed and resuspended in sterile 1× phosphate-buffered saline (PBS). The Mtb Erdman was then processed to disaggregate any clumps by vortexing five times with 3-mm sterile glass beads at 3 min intervals. *Mtb* Erdman suspension was then filtered using a 5-μm syringe filter to remove any remaining bacterial aggregations. The single cell suspension of now processed *Mtb* was then serially diluted and plated on 7H11 agar (Hi Media, Santa Maria, CA, USA) to determine the bacterial concentration of the processed stock. Aliquots of processed bacterial stocks were then frozen and stored in the cryogenic freezer at −80 °C. At the time of the experimental trial, the frozen-processed stocks of *Mtb* were thawed and used for the infection. 

### 4.3. Subject Recruitment

Study subjects were recruited after obtaining signed informed consent. The protocols for all the studies pertaining to in vitro *Mtb* infection were approved by the Institutional Biosafety committee of Western University of Health Sciences. The protocols for the study involving healthy subjects and participants with T2DM were approved by the Institutional Review Board of Western University of Health Sciences. Healthy adults (*N* = 8, ages 21 to 28 years, no known diseases or medication use) and individuals with T2DM (*N* = 5, ages 51 to 74 years, hemoglobin A1C > 7%) were recruited after obtaining signed informed consent. Exclusion criteria included the use of MFS within the last 6 months, or history of chemotherapy treatment within the last year. Other exclusion criteria included women who were currently pregnant, lactating, or had been pregnant within the last 6 months; pregnancy was considered a reason for study termination. After signing the informed consent form, 40 mL of blood was drawn from each participant.

### 4.4. Flavonoids

The mixed flavonoid supplement (MFS) was prepared by Reoxcyn LLC (Pleasant Grove, UT, USA). Supplement ingredients (US Patent 9,839,624) provided (in one tablet) 82.3 mg total monomeric flavonoids and included 25 mg vitamin C (as ascorbyl palmitate) (Green Wave Ingredients, La Mirada, CA, USA), wild bilberry fruit extract with 16 mg anthocyanins

(FutureCeuticals, Momence, IL, USA), green tea leaf extract with 46 mg total flavan-3-ols (Watson Industries, Inc., Pomona, CA, USA), 26 mg quercetin aglycone (Novel Ingredients, East Hanover, NJ, USA), 26.8 mg caffeine (Creative Compounds, Scott City, MO, USA), and 15 mg omega 3 fatty acids (Novotech Nutraceuticals, Ventura, CA, USA). As previously reported, the tablet contents were analyzed prior to the study for flavonoid content using high-performance liquid chromatography (HPLC) [22]. Thirteen anthocyanins were identified in the bilberry extract including delphinidin, cyanidin, petunidin, peonidin, and malvidin galactosides, glucosides, and arabinsosides. The supplement tablets were analyzed again after 12 weeks and 12 months of storage at room temperature, and the data indicate that all chemical components were stable. As previously described, vitamin C, caffeine, and omega 3 fatty acids were included as adjuvants to improve flavonoid bioactivity [22]. For in vitro administration, the flavonoid tablet (138 mg) was dissolved in 1 mL of 100% DMSO. Once dissolved, the solution was diluted with 9 mL of RPMI. This method brought the stock concentration to 13.8 mg of dissolved flavonoid tablet per mL. We used 25 uL of this solution per each 500 uL well, making the final concentration 0.69 mg of flavonoids per mL.

### 4.5. Determination of the Direct Antimycobacterial Effects of MFS

To determine the direct effects of MFS in altering the survival of *Mtb* Erdman, bacteria (6 × 10^4^ /well) were grown in 7H9 media in 24 well tissue culture plates (Corning, NY, USA) in the presence and absence of MFS up to 15 days. *Mtb* Erdman cultures were then either sham-treated (control) or treated with the MFS at a concentration of 0.69 mg/per mL. The MFS treatment groups received their MFS supplementation every 4 days until termination. *Mtb* cultures were maintained at 37 °C, with 5% CO_2_ until they were terminated at 8 and 15 days post-infection to determine the viability of *Mtb*. *Mtb* viability was ascertained by plating the diluted samples on 7H11 agar medium (Hi Media, Santa Maria, CA, USA) enriched with albumin dextrose complex (ADC) (Gemini, West Sacramento, CA, USA).

### 4.6. Isolation of Plasma and Monocytes

Plasma and peripheral blood mononuclear cells (PBMCs) were isolated from the whole blood of T2DM subjects and healthy individuals by Ficoll-Paque (Sigma) density centrifugation. This procedure involves centrifugation of blood layered on Ficoll-Paque at a 1:1 ratio at 1800 rpm for 30 min. Plasma (the top layer) was collected and stored at −80 °C, while PBMCs (the second layer from the top) were further washed three times with 1× phosphate-buffered saline (PBS) from Fisher Scientific International Inc. (pH 7.4 ± 0.1) and then resuspended in RPMI containing L-glutamine and 5% human AB serum. 

### 4.7. Induction of Granulomas

PBMCs were plated on a 24-well tissue culture plate, precoated with 0.001% poly-l-lysine, and infected with *Mtb* at a multiplicity of infection (MOI) of 0.1:1 bacteria-to-cell ratio (approximately 6 × 10^4^ bacteria were added to 6 × 10^5^ PBMCs). Infected cells were maintained at 37 °C with 5% CO_2_ for up to 15 days. Infected PBMCs were then either sham-treated or treated with 0.69 mg of MFS every 4 days until termination. Granulomas were terminated at 8 days and 15 days post-infection to determine the effects of the MFS in altering the granulomatous responses against *Mtb* infection.

### 4.8. Cell Termination and Determination of Mtb Erdman Survival

Cell free supernatants from each well were first collected and stored, the cells remaining were then harvested by adding 250 μL of ice-cold, sterile 1× PBS (Sigma, St. Louis, MO, USA) followed by gentle scraping to achieve maximum recovery of cell lysate from the wells. Lysates were then efficaciously vortexed followed by a freeze/thaw cycle in order to ensure sufficient rupture of cells and release of intracellular *Mtb*. The collected lysates and supernatants were then diluted as necessary in sterile 1× PBS (Sigma, St. Louis, MO, USA) and plated on 7H11 agar medium (Hi Media, Santa Maria, CA, USA) enriched with ADC (GEMINI, West Sacramento, CA, USA) and incubated at 37 °C for 3 weeks in order to evaluate the mycobacterial growth or survival subsequent to MFS treatment by counting the colony forming units (CFUs).

### 4.9. Hematoxylin and Eosin Staining

Granulomas on cover-glasses terminated at 15 days post-infection were fixed with 4% Paraformaldehyde (PFA) at room temperature for an hour. Fixed granulomas were washed once with 1x PBS and stained with Rapid H&E (Scientific R&D Corp, NY, USA) for 2 min and the excess stain was washed away with tap water. The cover glasses were inverted and mounted onto slides with mounting media (HistoChoice, Solon, OH, USA).

### 4.10. Quantification of GSH Levels

Measurement of total and oxidized glutathione was performed using the GSH colorimetric assay kit from Arbor Assays (Ann Arbor, MI, USA). Granuloma lysates were first comprehensively mixed with an equal volume of cold 5% sulfosalicylic acid (SSA), followed by incubation for 10 min at 4 °C, and subsequently centrifuged at 14,000 rpm for 10 min. The GSH was thereupon measured in the supernatants following the manufacturer’s instructions. Reduced GSH (rGSH) was calculated by subtracting the oxidized glutathione disulfide (GSSG) from the total GSH. All measurements were corrected for total protein levels.

### 4.11. Quantification of Total Protein Levels

Total protein was measured with a BCA Protein Assay Kit and performed per user instructions from Thermo Scientific (Rockford, IL, USA).

### 4.12. Assay of Cytokines

Levels of IL-10, IL-12, and IFN-γ, from the supernatants of THP-1 cells, and granulomas derived from PBMCs of healthy subjects and individuals with T2DM were measured by sandwich ELISA. The assay kits were purchased from eBioscience (San Diego, CA, USA) and performed as per the manufacturers’ protocol.

### 4.13. Statistical Analysis

Statistical data analyses were performed using GraphPad Prism version 7 (San Diego, CA, USA). Baseline levels of GSH, rGSH, MDA, IL-6, IL-10, IL-12, IFN-γ, and TNF-α were compared between healthy individuals and T2DM group using the unpaired *t*-test with Welch correction. Reported values are in means. Statistical significance was determined at * *p* < 0.05, ** *p* < 0.005, *** *p* < 0.0005.

## 5. Conclusions

Collectively, the results demonstrate that MFS treatment strongly influences against *Mtb* infectivity, including antimycobacterial effects, GSH enrichment, cytokine regulation and augmented granuloma formation. Our results indicated that these benefits can be seen among different cell types including THP-1 macrophages and PBMCs of both healthy participants and immunocompromised individuals with T2DM. The countermeasure effect of MFS treatment against *Mtb* infectivity in PBMCs from study participants with T2DM was robust but did not achieve the low post-treatment levels attained with the health participants. The cell culture data presented in this article extends the current research of flavonoid’s prophylactic potentiality, and suggests that increased flavonoid intake may be attractive adjunctive strategy for managing TB. Although, the in-vivo relevance of data indicating anti-pathogenic activity of flavonoid glycosides and aglycones in cell cultures has not yet been clearly established. The independent antipathogenic effect of the nutrient mix included in the study supplement (vitamin C, caffeine, omega 3 fatty acids) was not measured in this study.

## Figures and Tables

**Figure 1 molecules-24-00851-f001:**
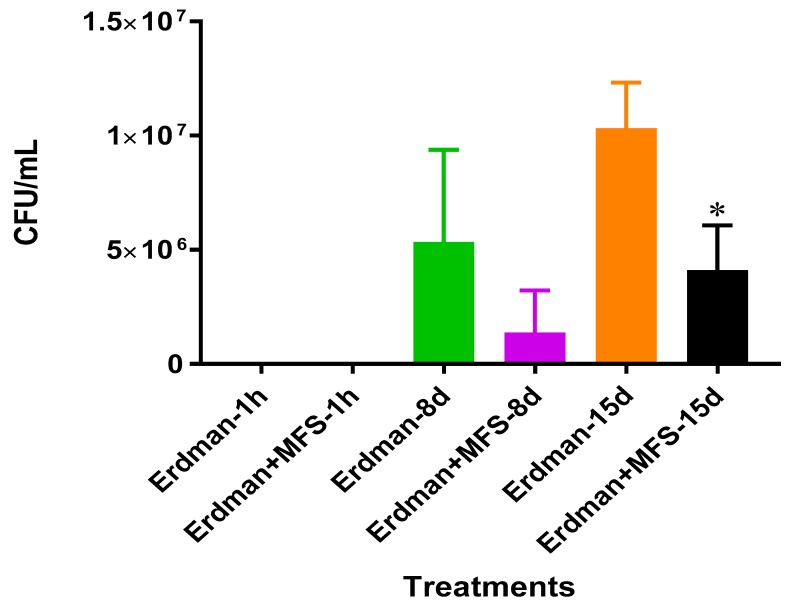
Growth of *Mtb* Erdman in 7H9 containing MFS. Experiments were performed in order to determine direct mycobacterial effects of each treatment. Data represent means ±SE from 1 trial and plating it multiple times. * *p* < 0.05 when comparing samples to their respective controls.

**Figure 2 molecules-24-00851-f002:**
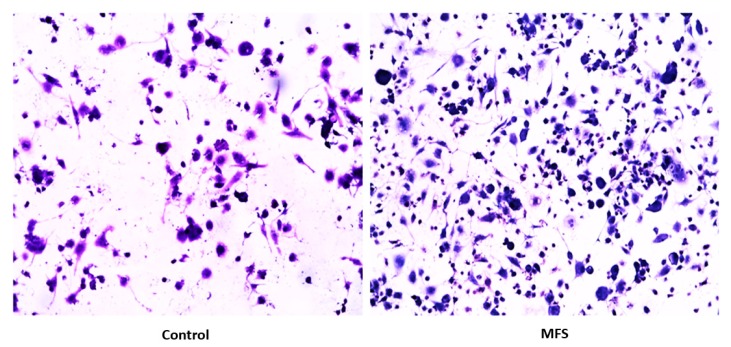
Hematoxylin and Eosin staining of untreated and MFS-treated samples of THP-1 cells. Microscopy work was done under 100×.

**Figure 3 molecules-24-00851-f003:**
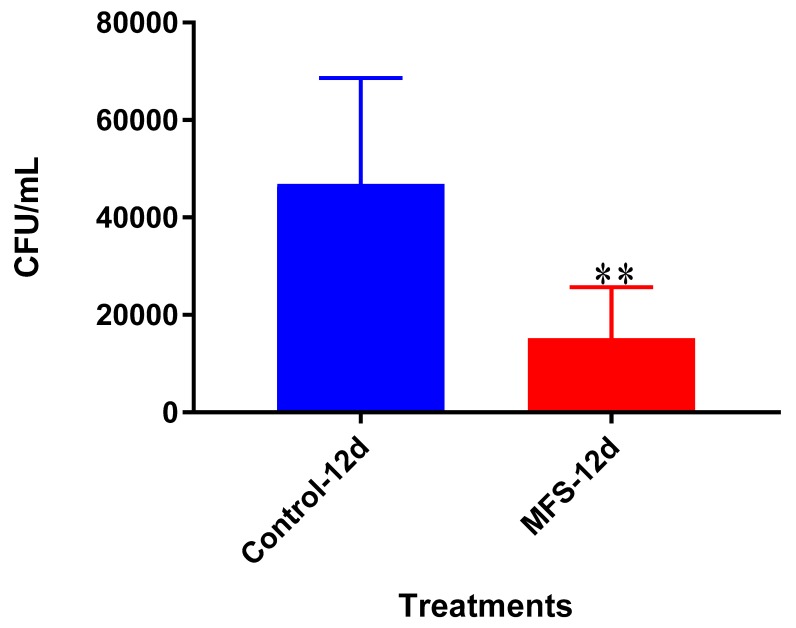
Survival of Mtb Erdman inside untreated and MFS-treated THP-1 macrophages. Data represent means ±SE from 6 trials and multiple plating times. ** *p* < 0.005 when comparing MFS-treated samples to untreated samples at 12 days.

**Figure 4 molecules-24-00851-f004:**
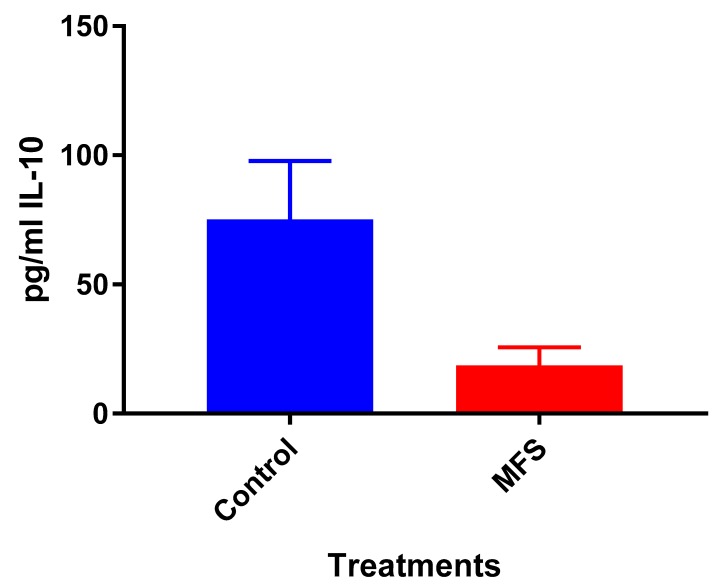
Levels of IL-10 in untreated and MFS-treated THP-1 cells. Assay of IL-10 was performed using an enzyme-linked immunosorbent assay (ELISA) Ready-Set-Go kit from eBioscience. Data represent means ±SE from 6 trials.

**Figure 5 molecules-24-00851-f005:**
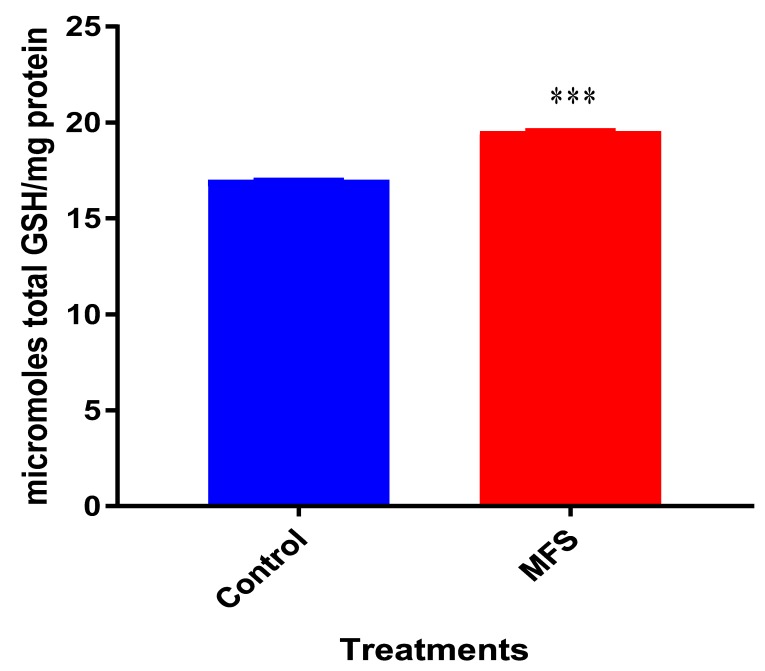
Levels of total Glutathione (GSH) in untreated and MFS-treated THP-1 cells. Corrections were made to total protein measured by BCA Protein Assay Kit from Thermo Scientific. Data represent means ±SE from 6 trials. *** *p* < 0.0005 when comparing GSH levels from control and MFS treated groups.

**Figure 6 molecules-24-00851-f006:**
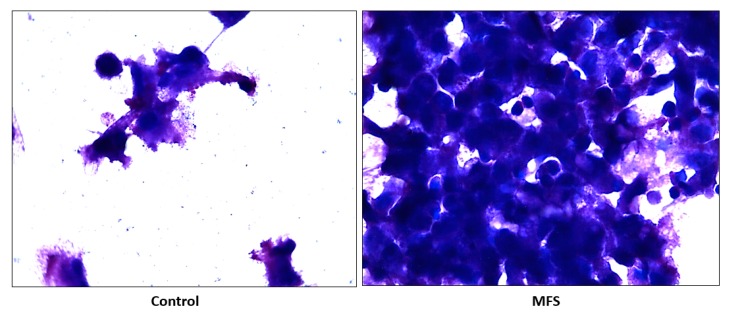
Hematoxylin and Eosin staining of untreated and MFS-treated human granulomas from healthy subjects. Microscopy work was done with a light microscope at 100× magnification under oil immersion.

**Figure 7 molecules-24-00851-f007:**
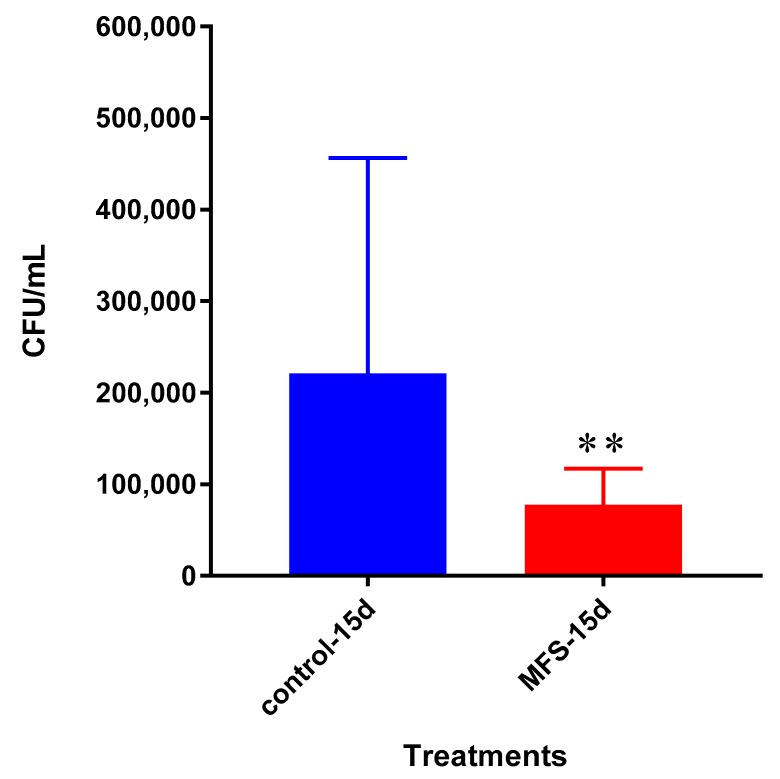
Survival of *Mtb* Erdman inside untreated and MFS-treated granulomas from healthy subjects. Data represent means ±SE from eight healthy individuals. ** *p* < 0.005 when comparing samples at 15 days.

**Figure 8 molecules-24-00851-f008:**
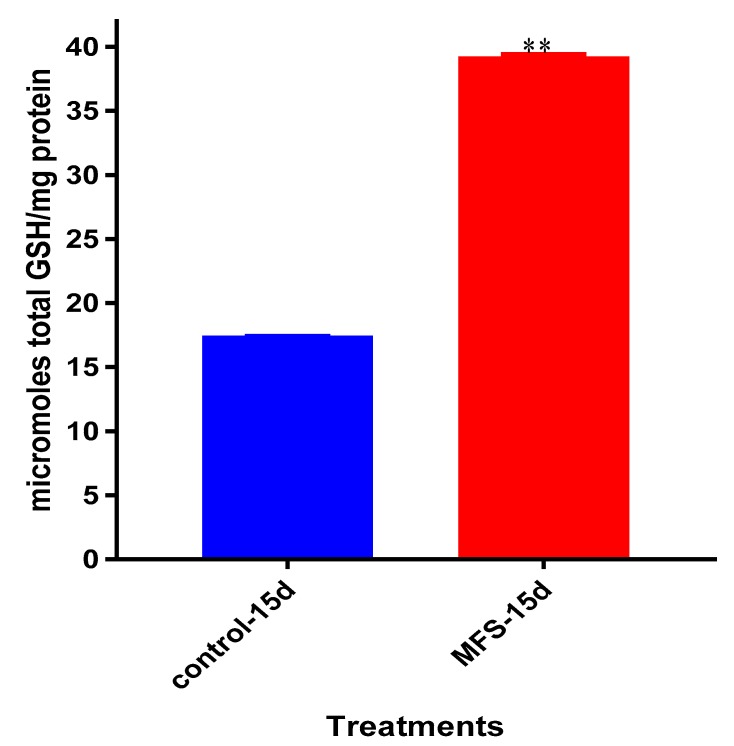
Levels of total GSH in untreated and MFS-treated granulomas from healthy subjects. Data represent means ±SE from eight healthy individuals. ** *p* < 0.005 when comparing GSH levels in control and MFS-treated groups.

**Figure 9 molecules-24-00851-f009:**
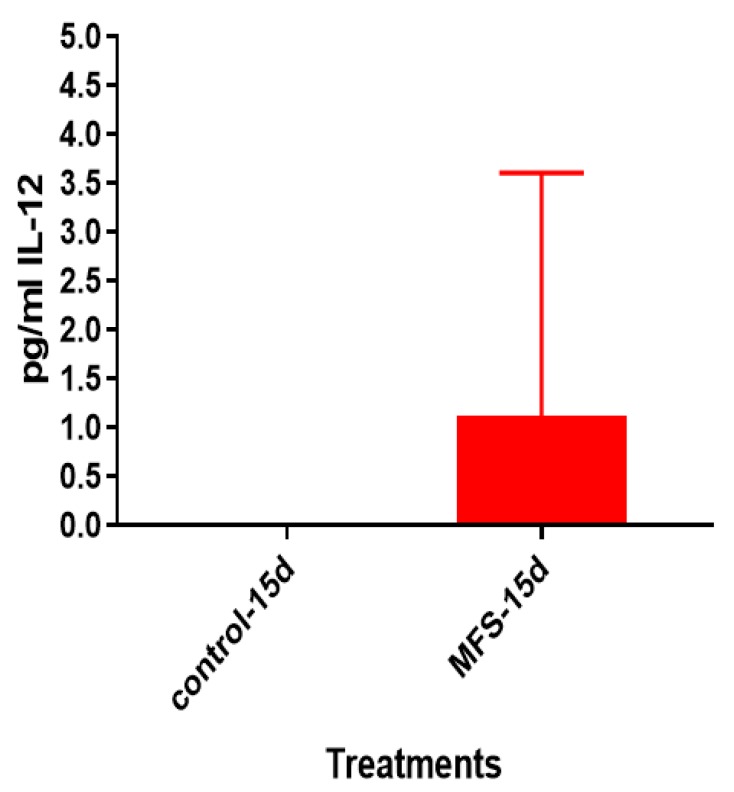
Assay of IL-12 in the supernatants from untreated and MFS-treated granulomas from healthy subjects. Assay of IL-12 was performed using an ELISA Ready-Set-Go kit from eBioscience. Data represent means ±SE from eight healthy individuals.

**Figure 10 molecules-24-00851-f010:**
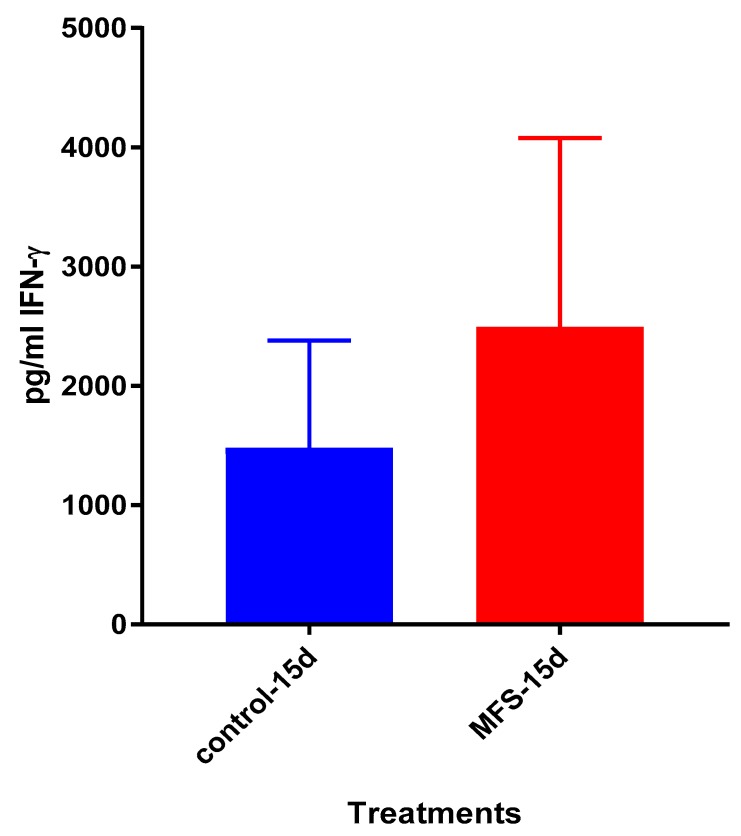
Assay of IFN-γ in the supernatants from untreated and MFS-treated granulomas from healthy subjects. Assay of IFN-γ was performed using an ELISA Ready-Set-Go kit from eBioscience. Data represent means ±SE from eight healthy individuals.

**Figure 11 molecules-24-00851-f011:**
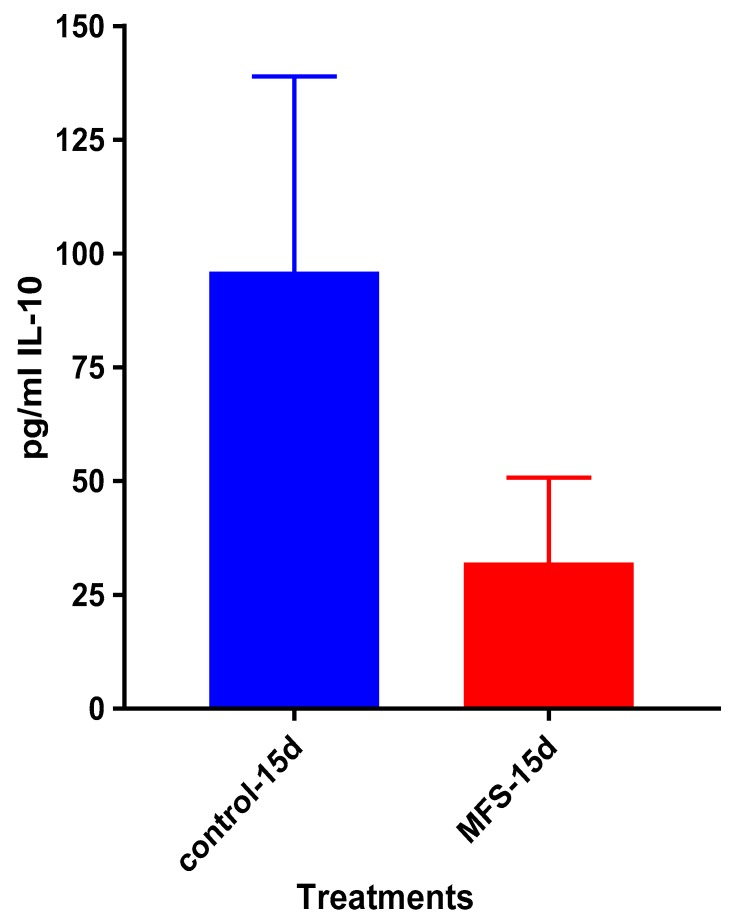
Assay of IL-10 in the supernatants from untreated and MFS-treated granulomas from healthy subjects. Assay of IL-10 was performed using an ELISA Ready-Set-Go kit from eBioscience. Data represent means ±SE from eight healthy individuals.

**Figure 12 molecules-24-00851-f012:**
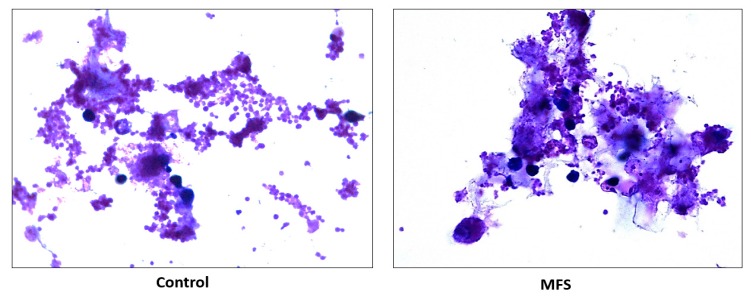
Hematoxylin and Eosin staining of untreated and MFS-treated human granulomas from individuals with T2DM. Microscopy work was done with a light microscope at 100× magnification under oil immersion.

**Figure 13 molecules-24-00851-f013:**
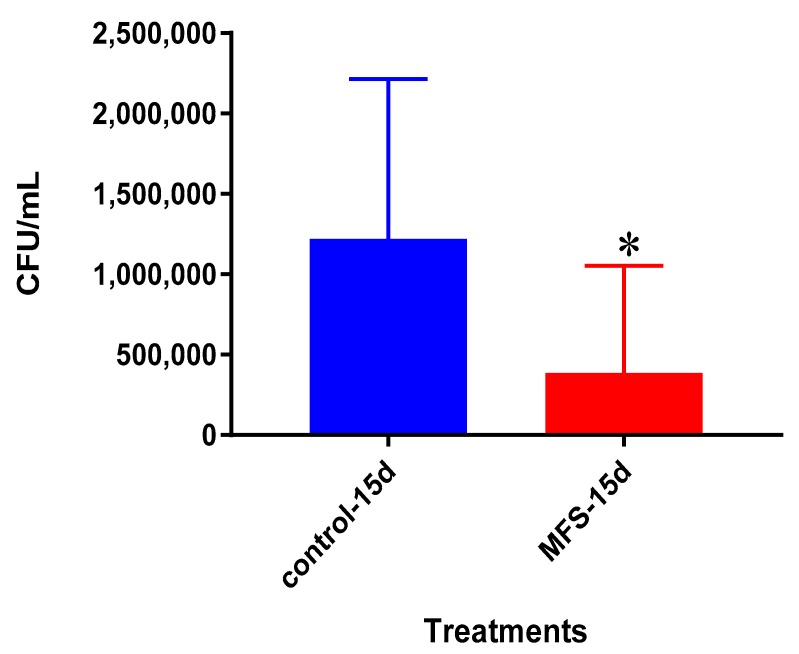
Survival *Mtb* Erdman inside untreated and MFS-treated granulomas from individuals with T2DM. Data represent means ±SE from six T2DM individuals. * *p* < 0.05 when comparing untreated samples with samples treated with MFS at 15 days.

**Figure 14 molecules-24-00851-f014:**
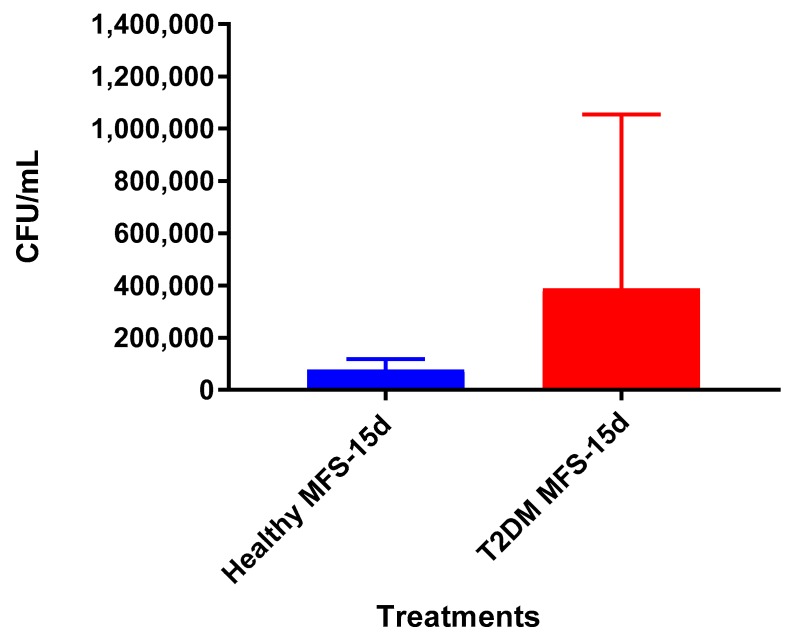
Survival of *Mtb* Erdman inside MFS-treated granulomas from healthy subjects and individuals with T2DM. Data represent means ±SE from six T2DM individuals and ±SE from healthy individuals.

**Figure 15 molecules-24-00851-f015:**
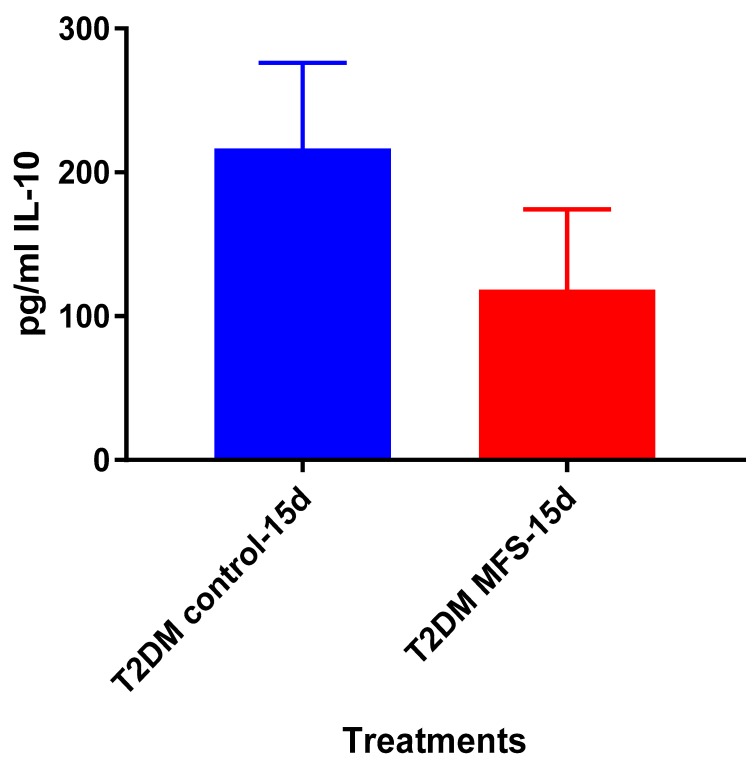
Levels of IL-10 in untreated and MFS-treated granulomas from individuals with T2DM. Assay of IL-10 was performed using an ELISA Ready-Set-Go kit from eBioscience. Data represent means ±SE from six T2DM individuals.

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
