# Peer review of "Flavonoid Mixture Inhibits *Mycobacterium tuberculosis* Survival and Infectivity"

_molecules, 2019, doi:10.3390/molecules24050851_

Round 1
Reviewer 1 Report
I have carefully read and reviewed the manuscript. Although it could be an interesting piece of science in showing the potential of natural products in positively influencing resolution of TB infection, results and experiments are poorly presented. Moreover, due to lack of some methods description, it is impossible to evaluate the activity/potential of the tested mixture.
Major points:
1) The authors state that the goal of this research work is to measure “…..the influence of flavonoids mixture on cell death and resolution of infection in a variety of M. tb infected cells.” (pg. 5 lines 101-102), but they test a mixture that contains caffeine, vitamin C and omega 3 fatty acids other than quercetin, green tea flavan-3-ols, and bilberry anthocyanins. Are these other molecules completely inactive? Why does the mixture contain other active ingredients?
2) Pg 10, lines 203-207, pg 13 lines 276-283. Measurement of Mtb survival. Due to the lack of the description of the experiment it is impossible to evaluate the potential in inhibiting bacterial growth. More specifically, how much was the inoculum? One needs that number to compare the effective growth inhibition. How did they measure the bacterial growth? Moreover, after 15 days of treatment bacteria started growing again compared to 8 days of treatment.
3) Pg 10, lines 215-219. Measurement of intracellular Mtb survival. Again, due to the lack of the description of the experiment it is impossible to evaluate the potential in inhibiting bacterial growth. More specifically, how much was the inoculum? One needs that number to compare the effective growth inhibition. How did they measure the bacterial growth?
4) Granulomas generation. The model used in this paper is not a reliable one.
In the light of above, the manuscript needs major revisions and it is not ready for publication, yet.
Minor points:
1) Common abbreviations for M. tuberculosis are either Mtb or MTB
2) Check references
3) Pg 10 line 205 “….M.tb growth in 7H9 media” pg 13 line 279 “….M. tb suspended in 7H11 broth media”. Which one is the one?
Author Response
Reviewer 1:
I would like to thank the reviewer for their thorough review to enhance the quality of the manuscript. We have made the alterations requested to resolve the specified concerns listed. Additionally, we are grateful and recognize the suggestions the reviewer has postulated and appreciate that more elucidation and clarification was needed for the specified areas mentioned. We have revised the manuscript to include all the changes that were recommended. The changes in the revised manuscript are denoted as track changes. Please see the specific responses below to the individual comments received.
Major points:
1) The authors state that the goal of this research work is to measure “…..the influence of flavonoids mixture on cell death and resolution of infection in a variety of M. tb infected cells.” (pg. 5 lines 101-102), but they test a mixture that contains caffeine, vitamin C and omega 3 fatty acids other than quercetin, green tea flavan-3-ols, and bilberry anthocyanins. Are these other molecules completely inactive? Why does the mixture contain other active ingredients?
We added this statement to the flavonoid section of the methods:
As previously described, vitamin C, caffeine, and omega 3 fatty acids were included as adjuvants to improve flavonoid bioactivity [17].
2) Pg 10, lines 203-207, pg 13 lines 276-283. Measurement of Mtb survival. Due to the lack of the description of the experiment it is impossible to evaluate the potential in inhibiting bacterial growth. More specifically, how much was the inoculum? One needs that number to compare the effective growth inhibition. How did they measure the bacterial growth? Moreover, after 15 days of treatment bacteria started growing again compared to 8 days of treatment.
Thank you for pointing out this error, we had forgotten to include this information in the Methods section. This mistake has now been rectified and we have included the section “Determination of the direct antimycobacterial effects of MFS” into the methods section containing all the pertinent information necessary including the inoculum amount and growth determination method.
3) Pg 10, lines 215-219. Measurement of intracellular Mtb survival. Again, due to the lack of the description of the experiment it is impossible to evaluate the potential in inhibiting bacterial growth. More specifically, how much was the inoculum? One needs that number to compare the effective growth inhibition. How did they measure the bacterial growth?
Again, thank you for pointing out this error, we had forgotten to include this information in the Methods section as well. This mistake has now been remedied and we have included the section “Cell Termination and Mtb Survival Determination” into the methods section containing all the necessary information. Additionally, we have added the inoculum quantity to the “Induction of Granulomas” Section.
4) Granulomas generation. The model used in this paper is not a reliable one.
We acknowledge that our method of granuloma generation is not perfect, however we feel that it is an adequate way to mimic and ascertain the immune response to a Mtb infection. Our granuloma model not only contains the primary cell types present in a typical Mtb infection but the relative proximity these cells would be in relation to each other in vivo which may very likely alter cytokine and immune response quorum. We would also like to draw your attention to Kapoor et al’s paper titled Human granuloma in vitro model, for TB dormancy and resuscitation (PLoS One) in which a similar method of granuloma generation was used, and our labs recent publication The Synergistic Effects of the Glutathione Precursor, NAC and First-Line Antibiotics in the Granulomatous Response Against Mycobacterium tuberculosis (Front Immunol) where we also used this same method.
Minor points:
1) Common abbreviations for M. tuberculosis are either Mtb or MTB
Thank you for your elucidation, we have now edited the manuscript so that “M. tb” has been replaced with “Mtb” throughout the article.
2) Check references
We have rechecked our references so that they meet Molecules guidelines.
3) Pg 10 line 205 “….M.tb growth in 7H9 media” pg 13 line 279 “….M. tb suspended in 7H11 broth media”. Which one is the one?
Thank you for pointing out this inaccuracy. The paper now correctly reads, “adding MFS to Mtb suspended in 7H9 broth media.”
Reviewer 2 Report
The manuscript by Cao, et al. investigates the antimycobacterial properties of a commercially available flavonoid supplement (MFS). The research topic is relevant as new options to combat tuberculosis are urgently needed, and flavonoids are known to display a wide variety of antimicrobial properties.
Overall comments:
- As the impact of a flavonoid-rich supplement, containing quercetin, anthocyanins, and flavan-3-ols but also other ingredients, was studied, the title, Flavonoids Inhibit Mycobacterium tuberculosis Survival and Infectivity, is not appropriate for the manuscript in my opinion. Please consider revising the title to make it clear from the beginning that a flavonoid mixture was tested.
- Articles are missing here and there, and commas are used inconsistently.
- Abbreviation of Mycobacterium tuberculosis should be Mtb or MTB, not M. tb, please revise throughout the manuscript.
Specific comments:
Abstract
- lines 33-35: influence is used three times, too repetitive, please revise line 34
Introduction
- lines 62-64: antibiotics should not be capitalized since they are not the trade names, please revise
- line 62: disease is not needed after TB, please revise
- lines 75-78: the two sentences starting by type 2 diabetes… do not provide any extra value for the manuscript. Consider removing them.
- line 82: the first sentence is too vague, please revise
- lines 84-86: What is the Phenol-Explorer database? The sentence: The Phenol-Explorer database contains values for 500 different polyphenols in over 400 foods in the human diet does not describe the database at all. It is just directly copy-pasted from the website, please revise.
- line 88: replace the comma by a dot.
- lines 89-95: Do you refer to flavonoids or polyphenols in general? It remains unclear as the words polyphenols and flavonoids are used incoherently. The articles you have cited focus on flavonoids.
- lines 97-98: What kind of anti-pathogenic influence? The ref 14 does not support the sentence, please revise.
Materials and methods
- line 109: origin of THP-1 cell line?
- line 122: Erdmann strain of M. tb (will henceforth be referred to as M. tb) not true, please revise the text or this sentence and apply it consistently throughout the manuscript.
- line 123: Middlebrook 7H9 media? Also, please provide the supplier. ACD stands for what?
- line 131: italics
- lines 142-153: Flavonoids: What are the anthocyanins and flavan-3-ols included in the tablets? Even though those can be found in the paper you cite, they should be listed here as well.
- line 165: supplier and pH of PBS?
- line 170: spell out the abbreviation MOI
- line 187: write out the abbreviation GSSG
- lines 200-203: Statistical analysis: statistical significance (p-values)?
Discussion
- Can you clearly state that all the observed effects are only due to the flavonoid content of the tablet? Can you say something about the role of omega-3 fatty acids?
- line 282: What was the media used for the experiments? 7H9 or 7H11?
- line 333: …the ill effects of this disease on M. tb infectivity are not completely eradicated. Please revise.
- line 341: These cell culture data extend those published from other research groups, please revise.
References
- Must be revised, provide all the references in the same format.
- 5. Association, A.D., Diagnosis and classification of diabetes mellitus. Diabetes care, 2014. 37(Supplement 1): p. S81-S90. Please revise, not written by Association AD but American Diabetes Association.
Legends, figures
- Please revise all, lots of unnecessary information included. Results are discussed elsewhere, not here.
Author Response
Rebuttal to Reviewer# 2
Reviewer 2:
I would like to thank the reviewer for the thorough review to enhance the quality of the manuscript. We have made the alterations requested to resolve the specified concerns listed. Additionally, we are grateful and recognize the suggestions the reviewer has postulated and appreciate that more elucidation and clarification was needed for the specified areas mentioned. We have revised the manuscript to include all the changes that were recommended. The changes in the revised manuscript are denoted as track changes. Please see the specific responses below to the individual comments received.
Overall comments:
- As the impact of a flavonoid-rich supplement, containing quercetin, anthocyanins, and flavan-3-ols but also other ingredients, was studied, the title, Flavonoids Inhibit Mycobacterium tuberculosis Survival and Infectivity, is not appropriate for the manuscript in my opinion. Please consider revising the title to make it clear from the beginning that a flavonoid mixture was tested.
As per your request, we have renamed the manuscript “Flavonoid Mixture Inhibits Mycobacterium tuberculosis Survival and Infectivity” so that it is more unambiguous that a flavonoid mixture was tested.
- Articles are missing here and there, and commas are used inconsistently.
We have done our best to remedy this issue by adding pertinent articles for reference and removing commas to reduce inconsistency.
- Abbreviation of Mycobacterium tuberculosis should be Mtb or MTB, not M. tb, please revise throughout the manuscript.
Thank you for your elucidation, we have now edited the manuscript so that “M. tb” has been replaced with “Mtb” throughout the article.
Specific comments:
Abstract
- lines 33-35: influence is used three times, too repetitive, please revise line 34
We agree that the use of the word influence was too repetitive among these two sentences. Therefore, we have re written the sentences as follows, “Flavonoids have been shown to exert anti-pathogenic potentiality, but few studies have investigated their effects on Mycobacterium tuberculosis (Mtb) infectivity. We hypothesized that a flavonoid mixture would have a favorable influence on cell death and the resolution of Mtb infection in THP-1 macrophages and in granulomas derived from both healthy participants and those with type 2 diabetes mellitus (T2DM)” to minimize its use.
Introduction
- lines 62-64: antibiotics should not be capitalized since they are not the trade names, please revise
Thank you for pointing out this inaccuracy, we have now edited the manuscript so that the antibiotics are not capitalized.
- line 62: disease is not needed after TB, please revise
As per the reviewer’s recommendation, we have deleted the word “disease” after TB on line 62.
- lines 75-78: the two sentences starting by type 2 diabetes… do not provide any extra value for the manuscript. Consider removing them.
Per the reviewer’s appeal, we have removed the two sentences on lines 75-78 pertaining to T2DM to keep the manuscript more succinct.
- line 82: the first sentence is too vague, please revise
We agree that line 82 was too vague, therefore we have edited the sentence so that it now read, “Polyphenols are a structural class of organic plant-based compounds, characterized by the presence of large phenolic units and responsible for many natural food pigmentations.”
- lines 84-86: What is the Phenol-Explorer database? The sentence: The Phenol-Explorer database contains values for 500 different polyphenols in over 400 foods in the human diet does not describe the database at all. It is just directly copy-pasted from the website, please revise.
We added this statement in this sentence: …"on polyphenol content in foods…"
- line 88: replace the comma by a dot.
Respectfully, we were not sure what was intended by this comment as we could not find a comma on line 88 that should have been replace by a period. If this was in reference to 1,187 mg/day, we double checked the paper referenced and this is the correct amount.
- lines 89-95: Do you refer to flavonoids or polyphenols in general? It remains unclear as the words polyphenols and flavonoids are used incoherently. The articles you have cited focus on flavonoids.
As emphasized earlier in this paragraph, nearly half of the polyphenols are flavonoids. Some databases and studies focus on polyphenols (larger group) and others on the flavonoid subset. We tried to be very careful in using the proper term based on the reference.
- lines 97-98: What kind of anti-pathogenic influence? The ref 14 does not support the sentence, please revise.
Reference 14 showed that flavonoid metabolites in serum (collected from human athletes) exerted an improved defense against viral replication. So this reference supports the statement. In response to your statement we changed the reference notation to 14-16 to include the three studies described in the paragraph.
Materials and methods
- line 109: origin of THP-1 cell line?
As per your recommendation, we have now added the origin of the THP-1 cell line (ATTC) used throughout our experiments.
- line 122: Erdmann strain of M. tb (will henceforth be referred to as M. tb) not true, please revise the text or this sentence and apply it consistently throughout the manuscript.
We acknowledge that we were incorrect with the use of this statement. Therefore, we have removed this assertion from the manuscript and instead declared that the Erdman strain was used for all our experiments conducted in this study.
- line 123: Middlebrook 7H9 media? Also, please provide the supplier. ACD stands for what?
Per your request, we have specified in the manuscript that Middlebrook 7H9 was used, added the supplier and indicated what ADC stands for.
- line 131: italics
Thank you for noticing this error, Mtb has now been italicized on line 131.
- lines 142-153: Flavonoids: What are the anthocyanins and flavan-3-ols included in the tablets? Even though those can be found in the paper you cite, they should be listed here as well.
We prefer to direct the reader to the original reference because the information is presented there in table and figure formats.
- line 165: supplier and pH of PBS?
Per your request, we have now specified the Supplier and pH of the PBS used in all our experiments.
- line 170: spell out the abbreviation MOI
Multiplicity of Infection has now been denoted as the significance of MOI on line 170 of the manuscript.
- line 187: write out the abbreviation GSSG
The abbreviation of GSSG and rGSH have now been written out in the manuscript.
- lines 200-203: Statistical analysis: statistical significance (p-values)?
The p-values have now been added under the statistical analysis section of the manuscript to convey when statistical significance was determined.
Discussion
- Can you clearly state that all the observed effects are only due to the flavonoid content of the tablet? Can you say something about the role of omega-3 fatty acids?
In the first paragraph of the discussion, we added this statement (in response to your comment):
… although we cannot rule out the potential synergistic effects of the added adjuvants caffeine, vitamin C, and omega 3 fatty acids.
- line 282: What was the media used for the experiments? 7H9 or 7H11?
Thank you for pointing out this inaccuracy on line 282. The paper now correctly reads, “adding MFS to Mtb suspended in 7H9 broth media.”
- line 333: …the ill effects of this disease on M. tb infectivity are not completely eradicated. Please revise.
As per your recommendation, we have revised line 333 of the manuscript so that it now reads, “This demonstrates that while MFS supplementation reduced Mtb infectivity in both healthy individuals and those with T2DM, it is not sufficient to cause complete Mtb elimination.”
- line 341: These cell culture data extend those published from other research groups, please revise.
Per your request, we have rewritten line 341, the sentence now states, “The cell culture data presented in this article extends the current research of flavonoid’s prophylactic potentiality, and when combined with epidemiological data suggests that increased flavonoid intake is an attractive adjunctive strategy for managing TB.”
References
- Must be revised, provide all the references in the same format.
We have rechecked our references so that they are presented in the same format, and meet Molecules guidelines.
- 5. Association, A.D., Diagnosis and classification of diabetes mellitus. Diabetes care, 2014. 37(Supplement 1): p. S81-S90. Please revise, not written by Association AD but American Diabetes Association.
Thank you for pointing out this error, this reference has now been rectified in the manuscript.
Legends, figures
- Please revise all, lots of unnecessary information included. Results are discussed elsewhere, not here.
We acknowledge and agree that there was a lot of unnecessary information contained in the legends including reiterations from the results. We have therefore done our best to remove the unnecessary information from this section.
Round 2
Reviewer 2 Report
The authors have addressed most of the comments and suggestions I gave in the first review. However, I still have some comments on the manuscript:
Original comments and replies:
- line 82: the first sentence is too vague, please revise
We agree that line 82 was too vague, therefore we have edited the sentence so that it now read, “Polyphenols are a structural class of organic plant-based compounds, characterized by the presence of large phenolic units and responsible for many natural food pigmentations.
New comment: All the phenolic units are not large, hence, please revise, for example, …characterized by the presence of multiple phenolic units…
- line 88: replace the comma by a dot.
Respectfully, we were not sure what was intended by this comment as we could not find a comma on line 88 that should have been replace by a period. If this was in reference to 1,187 mg/day, we double checked the paper referenced and this is the correct amount.
New comment: use a dot instead of the comma, in other words 1.187 mg/day not 1,187 mg/day
- lines 97-98: What kind of anti-pathogenic influence? The ref 14 does not support the sentence, please revise.
Reference 14 showed that flavonoid metabolites in serum (collected from human athletes) exerted an improved defense against viral replication. So this reference supports the statement. In response to your statement we changed the reference notation to 14-16 to include the three studies described in the paragraph.
New comment: I agree with the authors that flavonoids were shown to display antiviral activity (Ref 14). However, one reference is not enough for the statement …support a strong anti-pathogenic influence from selected flavonoids… without any further clarification. You should include what kind of anti-pathogenic activity was shown in Ref 14. Refs 15 and 16, in turn, clearly refer to the antimycobacterial effects of flavonoids.
- lines 142-153: Flavonoids: What are the anthocyanins and flavan-3-ols included in the tablets? Even though those can be found in the paper you cite, they should be listed here as well.
- We prefer to direct the reader to the original reference because the information is presented there in table and figure formats.
New comment: If this is the case, you should at least give some examples of anthocyanins and flavan-3-ols present in the MFS.
Legends, figures
- Please revise all, lots of unnecessary information included. Results are discussed elsewhere, not here.
We acknowledge and agree that there was a lot of unnecessary information contained in the legends including reiterations from the results. We have therefore done our best to remove the unnecessary information from this section.
Please revise again. For example, FIG 2B … THP cells were cultured in a medium of RPMI and 10% FBS and allowed to differentiate into macrophages by addition of PMA at a concentration of 10 ng/ml. Mtb infected macrophages (2x105 CELLS (?)/well) were either untreated or treated with MFS. This information belongs to the materials and methods section (and it is already there) not here.
Others:
- line 34: … to exert anti-pathogenic potential, not potentiality
- Results:
- Please rephrase the subtitles (2.1, 2.3, 2.5, 2.7, 2.11) to make them clearer. For example, 2.1 Mtb Survival Subsequent to MFS Treatment à Survival of Mtb Erdman Subsequent to MFS Treatment.
- lines 154, 261 and 288: The bacterial quantity of the Erdman strain of MTb à The bacterial quantity of Mtb Erdman, please revise.
- Materials and methods:
- Please rephrase the subtitles (4.2, 4.7, 4.9, 4.10). For example, 4.2 Culture of Erdmann strain of Mtb à Culture of Mtb Erdman or Mtb Erdman Culture; 4.9 Quantifying GSH levels à Quantification of GSH levels
- lines 415, 417, 424: Erdmann strain of Mtb à Mtb Erdman
- line 466: treated with MFS (0.69 mg/per ml) please revise
- Conclusions:
- lines 544-546: The cell culture data presented in this article extends the current research of flavonoid’s prophylactic potentiality, and when combined with epidemiological data suggests that increased flavonoid intake is an attractive adjunctive strategy for managing TB [12-16]. The conclusion should stand alone without any references. Moreover, the sentence remains unclear, please revise.
- References should be reformatted.
- Legends: Growth of Erdman strain of Mtb à Growth of Mtb Erdman etc
Author Response
Reviewer 2:
I would like to thank the reviewer for the thorough review to enhance the quality of the manuscript. We have made the alterations requested to resolve the specified concerns listed. Additionally, we are grateful and recognize the suggestions the reviewer has postulated and appreciate that more elucidation and clarification was needed for the specified areas mentioned. We have revised the manuscript to include all the changes that were recommended. The changes in the revised manuscript are denoted as track changes. Please see the specific responses below to the individual comments received.
The authors have addressed most of the comments and suggestions I gave in the first review. However, I still have some comments on the manuscript:
Original comments and replies:
- line 82: the first sentence is too vague, please revise
We agree that line 82 was too vague, therefore we have edited the sentence so that it now read, “Polyphenols are a structural class of organic plant-based compounds, characterized by the presence of large phenolic units and responsible for many natural food pigmentations.
New comment: All the phenolic units are not large, hence, please revise, for example, …characterized by the presence of multiple phenolic units…
We have revised the manuscript so that it now reads, “Polyphenols are a structural class of organic plant-based compounds, characterized by the presence of multiple units and responsible for many natural food pigmentations.”
- line 88: replace the comma by a dot.
Respectfully, we were not sure what was intended by this comment as we could not find a comma on line 88 that should have been replace by a period. If this was in reference to 1,187 mg/day, we double checked the paper referenced and this is the correct amount.
New comment: use a dot instead of the comma, in other words 1.187 mg/day not 1,187 mg/day
Per your suggestion we have amended the manuscript so that it now states, “…the average dietary polyphenol intake has been estimated at 1.187 g/day…”
- lines 97-98: What kind of anti-pathogenic influence? The ref 14 does not support the sentence, please revise.
Reference 14 showed that flavonoid metabolites in serum (collected from human athletes) exerted an improved defense against viral replication. So this reference supports the statement. In response to your statement we changed the reference notation to 14-16 to include the three studies described in the paragraph.
New comment: I agree with the authors that flavonoids were shown to display antiviral activity (Ref 14). However, one reference is not enough for the statement …support a strong anti-pathogenic influence from selected flavonoids… without any further clarification. You should include what kind of anti-pathogenic activity was shown in Ref 14. Refs 15 and 16, in turn, clearly refer to the antimycobacterial effects of flavonoids.
Added this statement in to the last paragraph of the introduction in response to your comment.
For example, serum samples collected from athletes that contained metabolites from blueberry and green tea ingestion protected cells from killing by the vesicular stomatitis virus [14].
- lines 142-153: Flavonoids: What are the anthocyanins and flavan-3-ols included in the tablets? Even though those can be found in the paper you cite, they should be listed here as well.
- We prefer to direct the reader to the original reference because the information is presented there in table and figure formats.
New comment: If this is the case, you should at least give some examples of anthocyanins and flavan-3-ols present in the MFS.
Added this statement to section 4.4 (Flavonoids):
Thirteen anthocyanins were identified in the bilberry extract including delphinidin, cyanidin, petunidin, peonidin, and malvidin galactosides, glucosides, and arabinsosides. Measured flavan-3-ols included epigallocatechin gallate (EGCG), epicatechin (EC), and epicatechin gallate (ECG).
Legends, figures
- Please revise all, lots of unnecessary information included. Results are discussed elsewhere, not here.
We acknowledge and agree that there was a lot of unnecessary information contained in the legends including reiterations from the results. We have therefore done our best to remove the unnecessary information from this section.
Please revise again. For example, FIG 2B … THP cells were cultured in a medium of RPMI and 10% FBS and allowed to differentiate into macrophages by addition of PMA at a concentration of 10 ng/ml. Mtb infected macrophages (2x105 CELLS (?)/well) were either untreated or treated with MFS. This information belongs to the materials and methods section (and it is already there) not here.
In accordance with your suggestion to revise any unnecessary material from the legends section, we have done our best to again remove all reiterated information from this section.
Others:
- line 34: … to exert anti-pathogenic potential, not potentiality
We have edited the document so that it now states potential instead of potentiality on line 34.
- Results:
- Please rephrase the subtitles (2.1, 2.3, 2.5, 2.7, 2.11) to make them clearer. For example, 2.1 Mtb Survival Subsequent to MFS Treatment à Survival of Mtb Erdman Subsequent to MFS Treatment.
Per your suggestion, we have rephrased subtitles 2.1, 2.3, 2.5, 2.7, and 2.11 to make them clearer to the reader.
- lines 154, 261 and 288: The bacterial quantity of the Erdman strain of MTb à The bacterial quantity of Mtb Erdman, please revise.
We have revised the aforementioned lines so that they read as “The bacterial quantity of Mtb Erdman” instead of “The bacterial quantity of the Erdman strain of MTb.”
- Materials and methods:
- Please rephrase the subtitles (4.2, 4.7, 4.9, 4.10). For example, 4.2 Culture of Erdmann strain of Mtb à Culture of Mtb Erdman or Mtb Erdman Culture; 4.9 Quantifying GSH levels à Quantification of GSH levels
Per your recommendation, we have rephrased the subtitles you requested to the manner you specified.
- lines 415, 417, 424: Erdmann strain of Mtb à Mtb Erdman
“The Erdmann strain of Mtb” has now been reworded to “Mtb Erdman” throughout the document.
- line 466: treated with MFS (0.69 mg/per ml) please revise
We have rewritten line 466 so that the document now reads “treated with the MFS at a concentration of 0.69 mg/per ml” for clarification.
- lines 544-546: The cell culture data presented in this article extends the current research of flavonoid’s prophylactic potentiality, and when combined with epidemiological data suggests that increased flavonoid intake is an attractive adjunctive strategy for managing TB [12-16]. The conclusion should stand alone without any references. Moreover, the sentence remains unclear, please revise.
Thank you for your concern, we have therefore revised this last sentence (lines 544-546) so that there is no longer any need for references, and elucidated the context we were attempting to convey.
- References should be reformatted.
We formatted the references
- Legends: Growth of Erdman strain of Mtb à Growth of Mtb Erdman etc
“The Erdmann strain of Mtb” has now been reworded to “Mtb Erdman” throughout the document.
